# Engineering of Ni(OH)_2_ Modified Two-Dimensional ZnIn_2_S_4_ Heterostructure for Boosting Hydrogen Evolution under Visible Light Illumination

**DOI:** 10.3390/nano12060946

**Published:** 2022-03-13

**Authors:** Huan Wang, Baorui Shao, Yaodan Chi, Sa Lv, Chao Wang, Bo Li, Haibin Li, Yingui Li, Xiaotian Yang

**Affiliations:** 1Key Laboratory for Comprehensive Energy Saving of Cold Regions Architecture of Ministry of Education, Jilin Jianzhu University, Changchun 130118, China; chiyaodan@jlju.edu.cn (Y.C.); lvsa82@163.com (S.L.); wangchao@jlju.edu.cn (C.W.); 2Department of Materials Science, Jilin Jianzhu University, Changchun 130118, China; br13180828503@163.com (B.S.); q11833395833@163.com (B.L.); lhb980215@163.com (H.L.); l15584271740@163.com (Y.L.); 3Department of Chemistry, Jilin Normal University, Siping 136000, China

**Keywords:** photocatalytic hydrogen evolution, charge transfer, heterostructure

## Abstract

Developing efficient catalysts to produce clean fuel by using solar energy has long been the goal to mitigate the issue of traditional fossil fuel scarcity. In this work, we design a heterostructure photocatalyst by employing two green components, Ni(OH)_2_ and ZnIn_2_S_4_, for efficient photocatalytic H_2_ evolution under the illumination of visible light. After optimization, the obtained photocatalyst exhibits an H_2_ evolution rate at 0.52 mL h^−1^ (5 mg) (i.e., 4640 μmol h^−1^ g^−1^) under visible light illumination. Further investigations reveal that such superior activity is originated from the efficient charge separation due to the two-dimensional (2D) structure of ZnIn_2_S_4_ and existing high-quality heterojunction.

## 1. Introduction

Efficient utilization of solar energy for generating benign hydrogen fuel from water has long been viewed as an ideal tactic for solving issues of energy dilemma and environmental pollution. Although dramatic progress has been achieved in related research areas [1,2], it is still challenging to obtain catalysts that could meet requirements of wide absorption range, high activity, good stability, and low cost. As of now, to boost the photocatalytic activity of the catalyst, tremendous efforts have been paid to the elaborate design of visible-light-driven photocatalysts, such as noble-metal free metals, carbides, sulfides, phosphides, and their modified compounds [3,4,5,6]. Among these studies, zinc indium sulfide (ZnIn_2_S_4_) has obtained tremendous interest owing to its merits of proper bandgap (2.3–2.7 eV), low toxicity, and cost-effectiveness [7]. However, a ZnIn_2_S_4_-derived photocatalyst usually exhibits moderate activity for hydrogen photogeneration, which is supposed to be due to its fast photogenerated carrier recombination drawback. To address this, various strategies have been employed to prolong the lifetime of carriers of ZnIn_2_S_4_ for enhancing its activity, including loading noble-metal nanocrystals, doping other elements, morphology control, and construction of a heterojunction [8,9]. For example, Li et al. prepared Pt/ZnIn_2_S_4_ composites through a hydrothermal method combined with a light-induced deposition tactic. A significant increase in the photocatalytic hydrogen evolution performance of the composite was observed as expected by using ethanolamine as an electron donor [10]. Yao et al. synthesized oxygen-doping ZnIn_2_S_4_ ultrathin nanosheets via a hydrothermal method [11]. Results showed that the obtained oxygen-doped ZnIn_2_S_4_ nanosheets exhibit much enhanced photocatalytic activity under the illumination of visible light, in which they postulated that the increased performance is possibly attributed to the effective separation of photogenerated charge carriers on the surface of the catalyst. Similarly, Yu et al. reported a ZnIn_2_S_4_@CuInS_2_ microflower core-shell p-n heterojunction by a hydrothermal method, which could efficiently increase the charge separation efficiency and, therefore, boost the activity of photocatalytic hydrogen production [12]. Zhu et al. employed RGO as an electron acceptor and cocatalyst to modify a ZnIn_2_S_4_ sheet, and the relevant hydrogen photogeneration performance of the prepared RGO/ZnIn_2_S_4_ nanocomposite was significantly improved [13].

In addition, previous investigations indicate that various nickel-containing species, including NiO, Ni(OH)_2_, and Ni_3_B, could act as the cocatalyst for the efficient reaction of photocatalytic hydrogen production [14,15,16]. Among them, the heterojunction, such as Ni(OH)_2_/TiO_2_, Ni(OH)_2_/C_3_N_4_, and Ni(OH)_2_/CdS, could obviously increase its photocatalytic activity under visible light illumination, which was supposed to be attributed to the inhibition of the recombination of photogenerated carriers [17,18,19]. Despite the progress, a randomly designed heterojunction structure has greatly restricted the separation efficiency of photogenerated carriers. Therefore, it is of great importance to develop an effective strategy, which could significantly mitigate the low-charge separation efficiency. To reach this goal, a tactic of forming high-quality 2D/2D heterostructures, such as Ni_2_P/ZnIn_2_S_4_ and MoS_2_/ZnIn_2_S_4_, is proposed, which would greatly decrease the charge migration distance, and therefore, the corresponding probability of charge recombination could be largely inhibited [20,21]. This unique structure is composed of two different materials with a 2D layered structure; usually, one is as a light absorber, while the other is as a cocatalyst. Benefitting from the elaborate structural design, it possesses the merits of short diffusion distance, large interface contact area, and rich active sites, which are postulated to efficiently promote the charge separation and transfer property at the interface of the heterojunction and therefore further improve the relevant catalytic activity. Recently, preliminary attempts were made based on this concept, where the composite comprises Ni(OH)_2_ and ZnIn_2_S_4_; [22,23] however, a further insightful investigation is still highly needed.

In this paper, the 2D ZnIn_2_S_4_ nanoflakes modified by thin Ni(OH)_2_ nanosheets were simply prepared by employing a two-step solvothermal method. Our results demonstrate that obtained composite exhibits enhanced the performance for hydrogen photogeneration under the illumination of visible light under optimal conditions. Further, the plausible underlying mechanism is proposed accordingly.

## 2. Experimental Section

### 2.1. Synthesis of Ni(OH)_2_ Nanosheets

The synthesis was according to previous work [24]. Typically, 1 mmol Ni(NO_3_)_2_·6H_2_O was added into a beaker containing 20 mL ethanol under vigorous stirring. After ~10 min, 2 mL oleylamine in 10 mL ethanol was quickly added to the above solution. The obtained homogeneous solution was stirred for further 30 min and then transferred into a 50 mL Teflon-lined autoclave. The autoclave was then kept at 180 °C for 15 h, and after that, it was cooled to room temperature. The resulting green product was collected by centrifugation and washed repeatedly with cyclohexane, deionized (DI) water, and ethanol three times. Finally, the obtained product was put in a vacuum furnace at 60 °C for 6 h for further use.

### 2.2. Synthesis of ZnIn_2_S_4_/Ni(OH)_2_ 2D/2D Composite

Typically, a certain amount of Ni(OH)_2_ nanosheets was dispersed into 40 mL DI water with subsequent sonicating for 10 min to form a stable suspension. Then, the suspension was transferred to 100 mL of the flask containing 10 mL glycerin and magnetically stirred for 30 min. Subsequently, 272 mg of ZnCl_2_, 1172 mg of InCl_3_·4H_2_O, and 602 mg of thioacetamide (TAA) were added into the above flask and stirred for further 20 min. The resulting mixture was heated at 80 °C for 2 h in an oil bath with continuous stirring. The product was subjected to the centrifugation and washing (with ethanol) step three times to remove any unreacted reagents and side products and then dried at 60 °C for 6 h for further use. Depending on the weight content of Ni(OH)_2_, which was evaluated by the inductively coupled plasma optical emission spectrometer (ICP-OES), the as-synthesized sample was denoted as x wt% Ni(OH)_2_/ZnIn_2_S_4_, and the detailed results can be found in Appendix A. Pure ZnIn_2_S_4_ was also prepared as a control with a similar procedure except without introducing Ni(OH)_2_ nanosheets during the synthesis.

### 2.3. Characterization

The crystal structure of all samples was accomplished on an X-ray diffractometer (Rigaku D/Max 2550, Wilmington, MA, USA, Rigaku Co., Ltd.) with a Cu Kα radiation (λ = 0.154056 nm). The morphology, elemental composition, and energy dispersive X-ray (EDX) analysis of the as-prepared samples were characterized by FE-SEM (JSM-7610F) and TEM (Fei Tecnai G2 F20 S-TWIN). XPS spectra were recorded on an ESCALAB MKII photoelectron spectrometer with Al Ka X-ray radiation. UV–VIS diffuse reflectance spectra were determined on a Shimadzu UV-2600 spectrophotometer with BaSO_4_ as a reference. The Brunauer–Emmett–Teller (BET) surface area of the samples was measured by a Micromeritics ASAP 2020 instrument, and before the measurements, all the samples were subjected to the heating treatment under 120 °C and vacuum condition for 6 h (note: the heating treatment did not alert the crystal structure of the samples, Appendix A). Steady-state photoluminescence (PL) spectra and time-resolved transient PL decay spectra of the samples were carried out on an FLS-1000 fluorescence spectrophotometer. For steady-state PL measurements, the excitation wavelength is set to 480 nm, while for the transient PL decay spectra, the excitation and emission wavelength are set to 450 and 550 nm, respectively. The photocurrent was evaluated using the photoelectrochemical (PEC) cell with three electrodes at several on–off irradiation cycles. Electrochemical impedance spectroscopy (EIS) experiments were tested on a potentiostat (0.2 V) in the Na_2_SO_4_ (0.5 M) solution, with an Ag/AgCl reference electrode. Photoelectrodes used for the relevant measurements were employed FTO (fluorine-doped tin oxide) glass sheets (1.0 × 4.0 cm)as the conductive substrate, and the details of the preparation of electrode are as follows: First, the FTO electrode was successively cleaned with DI water, ethanol, and acetone by sonication, 15 min for each step. Then, a piece of tape was employed to cover the electrode, which left the exposed area fixed at 1.0 × 1.0 cm for further sample deposition. Next, 1.0 mg of relevant sample was dispersed into 0.5 mL of ethanol and subjected to sonication for 15 min. After that, 10 μL of the corresponding solution was taken and dropped onto the electrode for further measurements after it was dried under ambient conditions. ICP-OES of the samples was measured by a Thermo Scientific iCAP 6300.

### 2.4. Photocatalytic Reaction Measurements and Calculation

The photocatalytic hydrogen evolution reaction was carried out in a gas-tight glass flask (50 mL). Typically, 5 mg of the photocatalyst was dispersed into 15 mL DI water containing triethanolamine (TEOA) (20 vol%) as electron donors. Before the reaction, the system was evacuated and then filled with nitrogen for 5 and 30 min, respectively, to ensure the thorough elimination of residual oxygen in the system. A 300 W xenon lamp coupled with a filter (>420 nm) was used as the light source. The amount of hydrogen evolution was sampled (200 µL) from the headspace of the flask by a gas-tight syringe (Bonaduz, Switzerland, Hamilton) and immediately detected by gas chromatography (GC-2014c, Suzhou, China, Shimadzu) at given time intervals.

## 3. Results and Discussion

The crystal structure of different samples was obtained from XRD measurements. As shown in Figure 1, pure Ni(OH)_2_ exhibits the characteristic diffraction peaks at 2θ = 11.7°, 24.7°, 33.1°, 35.2°, 42.5°, and 59.3°, which are indexed to the (001), (002), (110), (111), (103), and (300) crystal planes of the hexagonal crystal structure of α-Ni(OH)_2_ (JCPDS card no. 22-0444) [25]. Interestingly, for composite samples, only peaks of ZnIn_2_S_4_ corresponding to 21.2° (006), 27.6° (102), 30.5° (104), 47.2° (110), 52.4° (116), and 55.8° (022) for planes of a hexagonal crystal structure (JCPDS No. 65-2023) were observed, while no peak of Ni(OH)_2_ could be detected, which is postulated to be ascribed to the low content of Ni(OH)_2_ existing in the samples [26].

Then, the structural information of the obtained samples was acquired by SEM and TEM measurements. As an introduced material, Ni(OH)_2_ exhibits the 2D nanoflake morphology with the in-plane size from 200 to 500 nm and the thickness at ~20 nm (Appendix A). Further coating of ZnIn_2_S_4_ As for the Ni(OH)_2_/ZnIn_2_S_4_ composite, taking the 0.37 wt% one, for example, SEM image reveals that it exhibits the 2D nanoflower-like morphology with a hierarchical structure consisting of plenty of ultrathin nanosheets (Figure 2a). However, it should be noted that such morphology is well in line with that of pure ZnIn_2_S_4_, probably owing to the low introducing content of Ni(OH)_2_.

Our results clearly demonstrate that the introduction of Ni(OH)_2_ nanoflake during the synthesis does not significantly alert the formation dynamic of ZnIn_2_S_4_ nanosheets. Next, TEM and HRTEM measurements were applied to get further detailed structural information of 0.37 wt% Ni(OH)_2_/ZnIn_2_S_4_ composite. As indicated (Figure 2b), a clear ultra-thin-layered nanostructure was observed for Ni(OH)_2_/ZnIn_2_S_4_ composite, which is in line with SEM results (Figure 2a). Further HRTEM investigation (Figure 2c) unambiguously shows the interfacial region of ZnIn_2_S_4_ and Ni(OH)_2_, and as indicated, the lattice fringes with spacing at 0.27 and 0.32 nm could be ascribed to (110) plane of hexagonal Ni(OH)_2_ and (102) plane of hexagonal ZnIn_2_S_4_, respectively [27,28,29]. No selected area electron diffraction (SEAD) signal of Ni(OH)_2_ was observed for 0.37 wt% Ni(OH)_2_/ZnIn_2_S_4_ composite compared with that of Ni(OH)_2_ (Appendix A), which is assumed to be due to the low amount of Ni(OH)_2_ in the composite. HAADF-STEM (Figure 2d) and the corresponding elemental mapping results (Figure 2e–h) revealed the homogeneous distribution of Zn, In, S, and Ni elements throughout the sample, strongly verifying the successful synthesis of the designed structure. EDX measurement (Appendix A) further verifies the existence of Ni, though its content is low.

To confirm the chemical state of different elements of the as-prepared samples, XPS measurements were further carried out. Figure 3a represents the XPS survey spectra of ZnIn_2_S_4_ and 0.37 wt% Ni(OH)_2_/ZnIn_2_S_4_ composite, which confirms the existence of the designated elements only except for Ni. Furthermore, as shown in Figure 3b, Zn 2p XPS spectra of ZnIn_2_S_4_ and Ni(OH)_2_/ZnIn_2_S_4_ composite exhibit two peaks at 1044.2 and 1021.1 eV, which correspond to Zn 2p_1/2_ and Zn 2p_3/2_ of ZnIn_2_S_4_, respectively, evidencing the existence of Zn^2+^ in the sample [30,31]. Peaks (Figure 3c) at 452.2 and 444.5 eV can be indexed to In 3d_3/2_ and In 3d_5/2_, confirming that element In in the sample is in the form of a trivalent cation [32]. In addition, the binding energies of S 2p peak (Figure 3d) were split into two peaks 2p_1/2_ at 162.4 and 2p_3/2_ at 161.1 eV, which was proved to be the S^2−^ typical characteristic in metal sulfides [33]. However, it should be mentioned here that no signal of Ni 2p was detected, which is considered to be due to the extremely low amount of Ni(OH)_2_ in the sample.

BET measurements were then applied to evaluate the surface area of different samples. As indicated (Figure 3e and Appendix A), with the increase in Ni(OH)_2_ in the composite, the surface area is gradually increased from 0.51 m^2^/g (ZnIn_2_S_4_) to 124.65 m^2^/g (0.74 wt% Ni(OH)_2_/ZnIn_2_S_4_), while the pore size seems not to follow that rule. In addition, all the samples exhibit type IV (Brunauer–Deming–Deming–Teller classification), and the shape of the three hysteresis loops is type H3, assumed to be related to the aggregation of particles [16,34].

The absorbance properties of the as-obtained samples were then investigated by UV–VIS diffuse reflection spectroscopy. As shown (Figure 3f), the absorption of Ni(OH)_2_ consists of two wide absorption bands at 390–500 nm and 600–800 nm, corresponding to the d-d transition of Ni [19,35]. On the other hand, for the composite samples, the absorption characteristic does not show an obvious difference with varying Ni(OH)_2_ content in our case (Appendix A). In addition, new weak absorbance in the range of 600 to 800 nm, when compared with that of ZnIn_2_S_4_, further confirms the existence of Ni(OH)_2_.

The H_2_ photogeneration performance of different samples was evaluated by using TEOA as a sacrificial reagent under visible light illumination (>420 nm). Figure 4a shows the photocatalytic H_2_ evolution versus illumination time of different samples. The results show that Ni(OH)_2_ does not give any photocatalytic activity, while pure ZnIn_2_S_4_ only exhibits a pretty low activity (0.13 mL h^−1^), which is assumed to be related to the fast recombination rate of charge carriers. Exceptionally, the 0.37 wt% Ni(OH)_2_/ZnIn_2_S_4_ composite exhibits much higher activity for H_2_ photogeneration, where obvious bubbles were observed after the reaction (Appendix A); however, the physical mixed control sample with the same content only shows moderate activity, which is only slightly higher than that of pure ZnIn_2_S_4_. Our results clearly demonstrate the importance of our strategy for obtaining the composite to achieve high performance of photocatalytic H_2_ evolution. Further optimizing the amount of Ni(OH)_2_ introduced in the composite reveals that content at 0.37 wt% gives the best performance of 0.52 mL h^−1^ (5 mg) (i.e., 4640 μmol h^−1^ g^−1^ (Figure 4b)), which is comparable with the recent benchmarking results (Appendix A). The long-term stability test of 0.37 wt% Ni(OH)_2_/ZnIn_2_S_4_ (Figure 4c) indicates good stability even after four cycles of photocatalytic reaction.

To elaborate on the underlying mechanism of this interesting activity enhancement, PL emission and lifetime spectra, time-resolved photocurrent, and electrochemical technique were employed. As indicated (Figure 5a), ZnIn_2_S_4_ exhibits a strong broadband PL emission in the range of 500–700 nm [36], while the obvious decrease in the PL intensity of samples after introducing Ni(OH)_2_ is observed, and the corresponding degree is increased with the increase in the introduced amount of Ni(OH)_2_, which is assumed to be attributed to the efficient photogenerated charge transfer from ZnIn_2_S_4_ to Ni(OH)_2_, thus decreasing the probability of emission relaxation of carriers in ZnIn_2_S_4_. The process is expected to be beneficial to the charge separation in the composite, correlating with the enhancement of the photocatalytic activity of the catalyst [37,38]. In addition, time-resolved photocurrent spectra (Figure 5b) indicate that all the composite samples exhibit higher response than pure ZnIn_2_S_4_ or Ni(OH)_2_, which strongly signifies the critical role of Ni(OH)_2_ for efficient charge separation.

The charge transfer property of different samples was further evaluated by electrochemical impedance spectroscopy (EIS). As indicated (Figure 5c), the smallest Nyquist plot demonstrates its fast charge transfer property of 0.37 wt% Ni(OH)_2_/ZnIn_2_S_4_ composite [39]. Further PL lifetime results (Figure 5d and Appendix A) of ZnIn_2_S_4_ and 0.37 wt% Ni(OH)_2_/ZnIn_2_S_4_ indicate that after the incorporation of the tiny amount of Ni(OH)_2_ into ZnIn_2_S_4_, the relevant average lifetime (τ_ave(PL)_) is decreased from 1.68 to 1.42 ns, unambiguously revealing the charge accelerating role of Ni(OH)_2_.

All of the above results intensely evidence that the boosting of the performance of the composite catalyst is highly possible, originated from the relatively high surface area and efficient charge carrier separation through the formation of the designed heterogeneous structure. Besides, it has been widely recognized that Ni(OH)_2_ could act as the co-catalyst to accept the photoinduced electrons and further complete a subsequent proton reduction reaction during the photocatalytic hydrogen evolution process [40,41,42]. Therefore, based on all of these results, a probable mechanism for the Ni(OH)_2_/ZnIn_2_S_4_ composite to boost the photocatalytic hydrogen evolution activity was proposed (Figure 1). Under the illumination of visible light, the electron is excited from the valence band of ZnIn_2_S_4_ into the conduction band, followed by the subsequent quick transfer into Ni(OH)_2_ to contribute to the proton reduction reaction. Since the position of the minimum conduction band of ZnIn_2_S_4_ is −1.35 V, which is more negative than the reduction potential of H^+^/H_2_ [43], the electron generated in ZnIn_2_S_4_ possesses the adequate ability to drive the proton reduction reaction. Meanwhile, the left hole would be consumed by TEOA, accompanied by the formation of the relevant oxidation products. Owing to the existence of Ni(OH)_2_, the electron photoinduced in ZnIn_2_S_4_ is effectively inhibited, which results in the significant enhancement of the photocatalytic activity of the relevant composite.

## 4. Conclusions

In this work, we designed and prepared a Ni(OH)_2_-modified 2D ZnIn_2_S_4_ heterogeneous photocatalyst to achieve the high performance of photocatalytic H_2_ evolution under visible light illumination. Benefiting its unique structure, under optimal conditions, the obtained sample exhibits superior activity for H_2_ photogeneration. Furthermore, the plausible underlying mechanism is also proposed after the detailed investigations. It is hoped that our tactic and obtained information could provide useful information for the future design of a high-performance photocatalyst.

## Data Availability

The data presented in this study are available on request from the corresponding author.

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
