# Peer review of "Engineering of Ni(OH)_2_ Modified Two-Dimensional ZnIn_2_S_4_ Heterostructure for Boosting Hydrogen Evolution under Visible Light Illumination"

_nanomaterials, 2022, doi:10.3390/nano12060946_

Round 1

Reviewer 1 Report

AA modified the manuscript accordind to the reviewer suggestion and now it is suitable for publication.

Author Response

Dear Editor,

We sincerely appreciate your kindness in offering an opportunity to revise our manuscript (ID: nanomaterials-1538519) and according to the reviewer’s constructive suggestions, we have carefully revised our manuscript trying to address all the concerns raised by the reviewer. As per your suggestion, all the modifications have been marked up by using the “Track Changes” function in the revised manuscript. Besides, the corresponding point-by-point response is also attached below. We do hope our manuscript now could meet the stringent standards of Nanomaterials. Thanks again for your time consumption and consideration.

Sincerely,

Huan Wang

Reviewer 2 Report

The Authors prepare a photocatalyst consisting of Ni(OH)2 and ZnIn2S4. They find photocatalytic activity is optimised when the weight loading is 0.37% of Ni(OH)2. They include a very brief discussion on the properties of the prepared photocatalysts and attempt to draw some conclusions on the possible reasons for the differences in photocatalytic performance they observe. Unfortunately, key experimental details are not provided and careful control experiments are not presented. The missing gaps make the work incomplete and currently not suitable for publication.

Specific comments:

  1. Characterisation should be completed on all samples shown in Fig 4b. This is of particular importance for methods where a large difference is observed - such as the nitrogen adsorption experiment. It is not as important for the XPS where the samples shown have identical spectra.
  2. The excitation wavelength for PL must be given, and the data must be presented with y-axis values if the intensity of the peaks is to be compared. Without values on the y-axis it cannot be said that a peak decreases in intensity.
  3. The photoelectrochemsitry - description of how the cell is made is vital. The graph requires values on the y-axis.
  4. Explain why 0.2 V vs Ag/AgCl was used as potential for EIS. Consider whether this is an appropriate voltage when looking at hydrogen evolution reaction. Ask an expert to look at these data.
  5. Calculate error for lifetime obtained from the PL lifetime decay. The two plots look very similar. Was this data obtained from a film or from solution? what was the excitation and emission wavelength?
  6. In the title, Ni(OH)2 is referred to as a cluster. I see no evidence that the Ni(OH)2 is a cluster. The SEM images show they are large plates.
  7. The major issue with these catalysts is their stability and longevity. This should be analysed.
  8. Ref 2 does not seem relevant
  9. Give number of repeats for data in Fig 4b
  10. The nitrogen adsorption experiments show a significant difference. Careful experiment should be completed to determine if the change is due to the varying amount of Ni(OH)2
  11. EDS maps in Figure 2 are likely false for Ni considering the low content of Ni. Along with the fact Ni was not observed in XPS. An EDS spectrum should be provided to show Ni (and other elements mapped) is clearly observed, or the Ni map should be removed.

Author Response

Dear Reviewer,

We sincerely appreciate your kindness in offering an opportunity to revise our manuscript (ID: nanomaterials-1538519) and according to the reviewer’s constructive suggestions, we have carefully revised our manuscript trying to address all the concerns raised by the reviewer. As per your suggestion, all the modifications have been marked up by using the “Track Changes” function in the revised manuscript. Besides, the corresponding point-by-point response is also attached below. We do hope our manuscript now could meet the stringent standards of Nanomaterials. Thanks again for your time consumption and consideration.

Sincerely,

Huan Wang

Reviewer 3 Report

The authors should check the attachment.

Author Response

(The authors gave the same response as above.)

Round 2

Reviewer 3 Report

Since the revised version is well-written in response to review comments and I am satisfied with it, this paper deserves to be published.